# Visual impairment and objectively measured physical activity and sedentary behaviour in US adolescents and adults: a cross-sectional study

Lee Smith,[1] Sarah E Jackson,[2] Shahina Pardhan,[3] Guillermo Felipe López-Sánchez,[4] Liang Hu,[5] Chao Cao,[6,7] Davy Vancampfort,[8,9] Ai Koyanagi,[10] Brendon Stubbs,[11] Joseph Firth,[12] Lin Yang[13]

LS and SEJ share first authorship

For numbered affiliations see end of article.

**Correspondence to**
Dr Sarah E Jackson;
s.e.jackson@ucl.ac.uk

## ABSTRACT

**Objectives** To compare levels of physical activity and sedentary time in a representative sample of US adolescents and adults with and without visual impairment.

**Design** Cross-sectional analyses were carried out using data from the National Health and Nutrition Examination Survey.

**Participants** The study population consisted of 6001 participants (adolescents n=1766, adults n=4235). The present analysis aggregated data from 2003 to 2004 and 2005–2006.

**Measures** Objective physical activity and sedentary behaviour assessment was conducted over 7 days. Distance visual acuity was measured for each eye in all participants 12 years and older. Participants' vision was categorised as: normal vision, uncorrected refractive error, non-refractive visual impairment. We estimated the sex-specific linear associations between presenting vision and objectively measured physical activity and sedentary patterns using adjusted generalised linear models in adolescents and adults.

**Results and conclusions** Adolescents with uncorrected refractive error and non-refractive visual impairment did not accumulate higher levels of sedentary time or lower levels of moderate-to-vigorous physical activity (MVPA) compared with those with normal vision. We observed no association between vision status and accelerometer measured MVPA in adults aged 20–49 years. We observed more time spent sedentary among females 20–49 years old with non-refractive visual impairment compared with those presenting normal vision (mean difference 329.8 min/week, 95% CI: 12.5 to 647.0). Adults 50 years and older with non-refractive visual impairment appeared to accumulate less lifestyle physical activity, particularly in women (mean difference −82.8 min/week, 95% CI: −147.8 to −17.8). Adult women with non-refractive visual impairment have lower levels of lifestyle physical activity and higher levels of sedentary time than those with normal vision. Taken together, these findings highlight the need for interventions to promote physical activity and reduce sedentary time in adult populations with visual impairment, specifically adult women.

## Strengths and limitations of this study

► Large, population-based sample of US adolescents and adults.
► Objective measures of physical activity and sedentary time.
► Analyses are of a cross-sectional design and thus it is not known whether visual impairment leads to low levels of activity and high levels of sedentary time or vice versa.
► Risk for developing diabetes and associated complications such as cataracts and diabetic retinopathy may be reduced in those with visual impairment by participating in adequate levels of physical activity and lower amounts of sedentary behaviour.

## INTRODUCTION

Physical activity may be defined as any bodily movement caused by contraction of skeletal muscle that results in energy expenditure[1] and may include activities such as structured exercise and sport, active travel (walking and cycling), occupational activity and household chores/gardening. Regular and sustained participation in physical activity is beneficial for almost every facet of health in children, adolescents and adults. For example, it has been associated with lower incidence of cardiovascular disease, cancer and osteoarthritis, and promotes positive mental health in all ages.[2] At the other end of the energy expenditure spectrum, excessive time in sedentary behaviour has been shown to be detrimental to both physical and mental health, and this is independent from physical activity levels.[3]

In light of the positive benefits of regular participation in physical activity and the detrimental effects of sedentary behaviour, guidelines have been developed. The WHO states that to maintain good health, adults

should achieve at least 30 min of moderate activity (eg, brisk walking) on five or more days of the week.[4] More-over, country-specific guidelines state that sedentary time should be kept to a minimum.[5–8] It is advised that children and adolescents should achieve 60 min of moderate activity on each day of the week and keep sedentary time to a minimum.[4] However, despite these recommendations, population levels of physical activity are low, particularly in Western countries.[9] Data from the National Health Nutrition and Examinations Survey (NHANES) showed that in 2005–2006 fewer than 10% of US adults met the recommended physical activity guidelines.[9] It is likely that any increase in energy expenditure is beneficial to health. It has been hypothesised that increases in physical activity energy expenditure may have health benefits regardless of how that increase is achieved (ie, achieved via short periods of moderate activity or long periods of light-intensity activity that may yield equivalent levels of energy expenditure).[10] Considering these potential health benefits, all levels of physical activity should be promoted in all populations.

Persons with disabilities have been shown to have low levels of physical activity and high levels of sedentary behaviour and understanding differences versus the general population is important.[11] The disability of reduced eyesight may be a key barrier to an active lifestyle in adults and adolescents. It has been suggested that in people with visual impairment, there is a lack of access to recreational and athletic programmes, and help or encouragement in developing suitable and safe physical recreation skills and habits.[12] Moreover, this population experiences activity limitations in walking, and environmental barriers such as transport and lack of accessible exercise equipment can hamper a person's ability to be physically active.[12 13] The authors of the present paper have shown in a sample of 6634 UK participants (mean age 65.0±9.2 years) those with poor vision were twice as likely to be physically inactive than those with good eyesight. Similar findings were found for the variable 'recognition of friends across street' and 'reading ordinary newspaper'.[14] The present authors have also found similar associations in young people.[15] These findings are of importance owing to a high prevalence of reduced eyesight. For example, in the USA, it has been estimated that approximately 14 million individuals aged 12 years or older have visual impairment (defined as distance visual acuity of 20/50 or worse).[16] However, key limitations to previously discussed analyses were that crude self-reported measures of vision were used (not allowing one to distinguish between types of eye conditions, refractive and non-refractive) and physical activity was self-reported. Self-reported physical activity is subject to bias owing to participants not being able to accurately recall physical activity and reporting higher levels of physical activity than the actual truth. Further research is needed using objective measures of vision and objective measures of physical activity behaviour such as accelerometers, which are a more accurate measure of free living physical activity.[17]

One previous study has looked at visual impairment and objective physical activity using the NHANES cohort. This study found that in those older than 20 years, individuals with normal sight took an average of 9964 steps per day and engaged in an average of 23.5 min per day of moderate-to-vigorous physical activity (MVPA), as compared with 9742 steps per day and 23.1 min per day of MVPA in individuals with uncorrected refractive error (p>0.50 for both) and 5992 steps per day and 9.3 min/day of MVPA in individuals with visual impairment (p<0.01 for both). In multivariable models, individuals with visual impairment took 26% fewer steps per day (p<0.01; 95% CI, 18% to 34%) and spent 48% less time in MVPA (p<0.01; 95% CI, 37% to 57%) than individuals with normal sight. However, this study did not look at associations between sedentary time, light physical activity and visual impairment, and did not explore levels of activity in adolescence or older adults.[18] Moreover, although this study controlled for the important covariates age, sex, race, obesity, education and systemic disease, it did not control for health behaviours such as smoking which has also previously been shown to be associated with both physical activity[19] and visual impairment.[20]

The present paper therefore aimed to compare levels of physical activity (light, MVPA, lifestyle) and sedentary time in a representative sample of US adolescents, adults and older adults using objective measures of physical activity and visual impairment controlling for a wide range of demographic and behavioural covariates.

## METHODS
### Study population
The NHANES was designed to provide cross-sectional estimates of the prevalence of health, nutrition and potential risk factors among the civilian non-institutionalised US population up to 85 years of age.[21] In brief, NHANES surveys a nationally representative complex, stratified, multistage, probability clustered sample of around 5000 participants each year in 15 counties across the country. Survey participants are asked to attend a physical examination either in a mobile examination centre (MEC) or in the participants' home. The present analysis aggregated data from 2003 to 2004 and 2005–2006. During these waves, objective physical activity and sedentary behaviour assessment was implemented in the NHANES participants by fitting them with a hip-worn accelerometer (ActiGraph AM-7164) for 7 days.

Data on sociodemographic information, measures of adiposity, smoking history, vision examination and objective physical activity were extracted and combined into a single dataset for each data collection wave, 2003–2004 and 2005–2006. Further, participants who were pregnant or had physical functional impairments that limited their ability to crawl, walk, run or play (age 12–19 years) or limited them from walking for a quarter mile, or walking up 10 steps (20 years and above) were excluded.

## Presenting visual acuity

NHANES participants undertook the vision examination at the MEC. The procedure of vision examination has been detailed elsewhere.[16] In brief, distance visual acuity was measured for all participants 12 years and older for each eye. An autorefractor (ARK0760, Nidek, Tokyo, Japan) was used which contains built-in visual acuity charts with 20/20, 20/25, 20/30, 20/40, 20/50, 20/60, 20/80 and 20/200 lines. Presenting visual acuity was defined using the better eyes with participants' usual distance version correction, if any. The 20/50 line was presented first, with at least four of the five characters to be read correctly to advance to the next line, otherwise the 20/200 line was presented. For eyes with presenting visual acuity of 20/30 or worse, visual acuity were measured after incorporating information from the objective refraction measurement. Participants with better seeing eyes with distance visual acuity of 20/30 or better were categorised as having normal vision. Participants with better-seeing eyes of presenting visual impairment that improved, aided by automated refraction result to 20/40 or better were categorised as having uncorrected refractive error, otherwise non-refractive visual impairment.[16]

## Accelerometer measured activity pattern

NHANES participants were asked during their physical examinations at the MEC to wear an accelerometer (Acti-Graph AM-7164, 1 min epochs) on the right hip for seven consecutive days to objectively measure free-living physical activity. The ActiGraph AM-7164 is a validated, small lightweight device that provides detailed information about the intensity, frequency and duration of physical activity.[22] The epoch length was set at 1 min, and the Actigraph recorded data for physical activity in the form of counts per minute (cpm). Non-wear time was defined as 60 min of consecutive zero counts. A recording of at least 10 hours of data was defined as a valid day, and four or more valid days were required to be included in the analysis. The total minutes of valid data were recorded as the accelerometer wear time. Based on standard cpm cut-off methods,[23] four raw activity outcomes were derived: sedentary behaviour (<100 cpm), light intensity physical activity (100–759 cpm), lifestyle activity or ambulatory (760–2109) and at least 10 min of MVPA (>2020 cpm). We further computed wear time adjusted activity by dividing each raw activity minutes by total wear time and multiplying the resulting fraction by the average wear time of all participants. We summarised the adjusted total weekly minutes of sedentary behaviour, light intensity physical activity, lifestyle physical activity and MVPA for each participant.

## Sociodemographic characteristics

Sociodemographic characteristics including age, sex, race and ethnicity, household income, employment status and smoking status were extracted. Based on self-reported race and ethnicity, participants were classified into one of the three racial groups: non-Hispanic white, non-Hispanic black and Hispanic and others. Annual household income was grouped into <US$20 000, US$20 000–US$74 999 and ≥US$75 000. Employment status was dichotomised to employed versus unemployed. We classified participants into three groups: never smokers (did not smoke 100 cigarettes and do not smoke now), former smokers (smoked 100 cigarettes in life and do not smoke now) and current smokers (smoked 100 cigarettes in life and smoke now).

## Overweight and obesity criteria

Weight and height were measured during the physical examination in MEC or in the participant's home for those whose travel was limited. The measurements followed standard procedures and were carried out by trained technicians with standardised equipment. Body mass index (BMI) was calculated as weight in kg/height in metres.[2] The standard definition for overweight and obesity classification was used to divide the BMI values into three categories: underweight or normal weight (<25.0), overweight (25.0–29.9) and obese (≥30.0).

## Chronic illness

We included four chronic conditions: cardiovascular diseases, diabetes, cancer and arthritis. Participants were considered as having chronic illness if they self-reported being told by a physician that they have the following conditions: congestive heart failure, coronary heart disease, heart attack, a stroke (cardiovascular diseases), diabetes, cancer or arthritis.

## Patient and public involvement

Patients and the public were not involved in the design of the present study.

## Statistical analysis

Survey analysis procedures were used to account for the sample weights, stratification and clustering of the complex sampling design to ensure nationally representative estimates. Descriptive statistics for participant characteristics were calculated by presenting vision. Sample size and weighted proportions were summarised.

Sex-specific linear associations between presenting vision (normal vision, uncorrected refractive error, non-refractive visual impairment) and objectively measured physical activity and sedentary patterns were estimated using generalised linear models in children and adolescents aged between 12 and 19 years, adults aged 20–49 years and older adults aged 50+ years, respectively. Generalised linear models were adjusted for age, race, BMI and household income among those between 12 and 19 years of age, and additionally adjusted for employment status, smoking and chronic illness among adults (20–49 years, and 50 years and older). Due to the small amount of light intensity physical activity and lifestyle physical activity in the younger population, we only included sedentary time and MVPA in models for population aged 12–19 years.

Finally, marginal means were estimated in multivariate adjusted generalised linear models for each modelled

outcomes of accelerometer measured activity pattern. All statistical significance was set at p<0.05. All statistical analyses were performed using Stata V.14.0.

## RESULTS

The study population consisted of 6001 participants who had data on presenting refractive error and accelerometer measured activity pattern. The majority of the participants were 20 years and older (n=4235). Overall, 60.8% (weighted proportion) of participants had presenting normal vision, 33.8% and 5.4% had uncorrected refractive error or non-refractive visual impairment. Participants who were older, female, unemployed, with education high school or lower and had a chronic condition were more likely to have non-refractive visual impairment compared with younger, male, employed,

**Table 1** Sociodemographic characteristics and objectively measured physical activity levels of the US population aged 12 years and older from the NHANES (2003–2006) by visual impairment

| | Normal vision | | Uncorrected refractive error | | Visual impairment | | |
|---|---|---|---|---|---|---|---|
| | N | Weighted % | N | Weighted % | N | Weighted % | P value |
| Overall | 3350 | 60.8 | 2225 | 33.8 | 426 | 5.4 | |
| Weighted N | 70 084 202 | | 38 937 983 | | 6 184 612 | | |
| Age group, years | | | | | | | 0.001 |
| 12–19 | 999 | 11.2 | 689 | 12.4 | 78 | 6.8 | |
| 20–49 | 1394 | 57.0 | 620 | 42.1 | 58 | 21.9 | |
| ≥50 | 937 | 31.8 | 916 | 44.5 | 290 | 71.3 | |
| Sex | | | | | | | 0.004 |
| Men | 1837 | 53.5 | 1114 | 47.0 | 223 | 49.9 | |
| Women | 1513 | 46.5 | 1111 | 53.0 | 203 | 50.1 | |
| Race/ethnicity | | | | | | | 0.257 |
| Non-Hispanic white | 1555 | 74.5 | 950 | 71.3 | 229 | 77.0 | |
| Non-Hispanic black | 802 | 9.6 | 543 | 10.7 | 86 | 9.5 | |
| Hispanic | 993 | 15.9 | 732 | 18.0 | 111 | 13.5 | |
| Other | 133 | 4.8 | 91 | 5.7 | 8 | 3.8 | |
| Household income | | | | | | | <0.001 |
| <20 000 | 964 | 17.5 | 806 | 24.7 | 209 | 38.6 | |
| 20 000–74 999 | 1740 | 52.2 | 1162 | 53.5 | 218 | 56.8 | |
| ≥75 000 | 976 | 36.5 | 511 | 31.6 | 56 | 18.0 | |
| Weight status* | | | | | | | 0.649 |
| Normal | 1416 | 37.1 | 989 | 39.4 | 171 | 37.5 | |
| Overweight | 1009 | 32.4 | 655 | 32.3 | 141 | 32.0 | |
| Obesity | 911 | 30.0 | 571 | 27.9 | 112 | 30.1 | |
| Employment status† | | | | | | | |
| Unemployed | 664 | 23.0 | 724 | 35.9 | 242 | 59.0 | |
| Employed | 1604 | 77.0 | 780 | 64.1 | 98 | 41.0 | |
| Smoking status† | | | | | | | 0.079 |
| Never smoker | 1243 | 46.0 | 780 | 45.0 | 167 | 44.6 | |
| Former smoker | 611 | 22.7 | 450 | 24.2 | 130 | 33.3 | |
| Current smoker | 496 | 20.0 | 306 | 18.4 | 51 | 15.3 | |
| Chronic condition‡ | | | | | | | <0.001 |
| Yes | 649 | 23.1 | 616 | 30.3 | 180 | 44.4 | |

*BMI was calculated as weight in kilograms divided by height in metres squared. Weight status was defined using BMI-to-age percentile in those aged 12–19 years, and WHO standard cut-off for adults 20 years and older.
†Data are only available on adults aged 20 years and older.
‡If participants have one of the following conditions: cardiovascular disease, diabetes, arthritis and cancer.
BMI, body mass index; NHANES, National Health Nutrition and Examinations Survey.

**Table 2** Accelerometer measured activity pattern (minutes per week) by vision status in the NHANES (2003–2006) children and adolescents aged 12–19 years

| | Male | | | Female | | |
|---|---|---|---|---|---|---|
| | Unadjusted β (95% CI) | MV-adjusted β (95% CI)* | Marginal mean | Unadjusted β (95% CI) | MV-adjusted β (95% CI)* | Marginal mean |
| *Sedentary time* | | | | | | |
| Normal vision | Ref | Ref | 3354.3 | Ref | Ref | 3541.5 |
| Uncorrected refractive error | −50.9 (−326.4 to 224.5) | −101.7 (−362.5 to 159.2) | 3252.6 | −63.3 (−257.0 to 130.4) | −41.4 (−226.85 to 144.1) | 3500.1 |
| Visual impairment | 697.5 (−326.4 to 1721.3) | 628.5 (−296.6 to 1553.5) | 3982.7 | 141.5 (−424.5 to 707.5) | 181.4 (−347.99 to 710.8) | 3722.9 |
| P trend | 0.578 | 0.815 | | 0.844 | 0.960 | |
| *Moderate-to-vigorous physical activity* | | | | | | |
| Normal vision | Ref | Ref | 104.8 | Ref | Ref | 47.7 |
| Uncorrected refractive error | −1.4 (−24.2 to 21.4) | −9.1 (−30.0 to 11.8) | 95.7 | −4.3 (−19.6 to 11.0) | −5.7 (−21.4 to 10.1) | 42.0 |
| Visual impairment | 63.9 (−1.2 to 129.1) | 51.3 (−16.4 to 119.0) | 156.1 | 10.2 (−36.9 to 57.2) | 7.7 (−36.7 to 52.1) | 55.4 |
| P trend | 0.432 | 0.870 | | 0.888 | 0.759 | |

*MV-adjusted models included age (continuous), BMI, race and household income.
BMI, body mass index; MV, multivariate; NHANES, National Health Nutrition and Examinations Survey.

well-educated participants and those with no chronic condition (table 1).

Tables 2–4 summarise both the non-adjusted and adjusted associations between presenting vision status and accelerometer measured activity pattern in younger (12–19 years old) participants and adults aged 20–49 years and ≥50 years, respectively. Children and adolescents aged between 12 and 19 years with uncorrected refractive error and non-refractive visual impairment did not accumulate higher levels of sedentary time or lower levels of MVPA compared with those with normal vision. The null association was seen in both males and females. After adjusting for age, BMI, race and household income, the estimated marginal mean of sedentary time in those with non-refractive visual impairment were 3982.7 min per week (equivalent 9.5 hours a day) and 3722.9 min per week (equivalent 8.8 hours a day) in males and females, respectively. The estimated marginal mean of MVPA in those with non-refractive visual impairment were 156 min per week in males and 55 min per week in females.

Similarly, no association was observed between presenting vision status and accelerometer measured MVPA in adults aged 20–49 years or those aged ≥50 years, although the multivariate marginal mean of MVPA among those aged ≥50 years with non-refractive visual impairment (48.1 min per week in male, 30.3 min per week in females) appeared lower than that in the young adult population (62.6 min per week in male, 49.8 min per week in females). With respect to sedentary behaviour, higher levels of sedentary time were observed among women 20–49 years with non-refractive visual impairment compared to those with presenting normal vision (mean difference min/week 329.8, 95% CI: 12.5 to 647.0). No association was observed between presenting vision and light intensity physical activity in either gender. However, adults 50 years and older with non-refractive visual impairment appeared to accumulate lower lifestyle physical activity, particularly in women (mean difference min/week −82.8, 95% CI: −147.8 to −17.8).

## DISCUSSION

In the present study of a large population based sample of the USA, we found that those aged 12–19 years with uncorrected refractive error and non-refractive visual impairment had similar levels of activity and sedentary time to those presenting with normal vision. Among adults aged 20–49 years, we found higher levels of sedentary time among women with non-refractive visual impairment compared to those with normal vision. Moreover, adults aged 50 years and older with non-refractive visual impairment appeared to accumulate lower lifestyle physical activity, particularly in women.

The finding that visually impaired adolescents (uncorrected refractive error and non-refractive visual impairment) exhibit little difference in their level of activity and sedentary behaviour compared with adolescents with 'normal' vision is interesting. To our knowledge, just one

**Table 3** Accelerometer measured activity pattern (minutes per week) by vision status in the NHANES (2003–2006) adults aged 20–49 years

| | Male | | | Female | | |
|---|---|---|---|---|---|---|
| | Unadjusted β (95% CI) | MV-adjusted β (95% CI)* | Marginal mean | Unadjusted β (95% CI) | MV-adjusted β (95% CI)* | Marginal mean |
| *Sedentary time* | | | | | | |
| Normal vision | Ref | Ref | 3367.7 | Ref | Ref | 3326.2 |
| Uncorrected refractive error | −55.8 (−213.0 to 101.4) | −13.5 (−180.8 to 153.9) | 3354.2 | −5.0 (−184.5 to 174.5) | 9.9 (−193.6 to 213.4) | 3336.1 |
| Visual impairment | 52.7 (−496.7 to 602.2) | 273.9 (−323.3 to 871.0) | 3641.5 | 257.9 (−90.8 to 606.5) | 329.8 (12.5 to 647.0) | 3655.9 |
| P trend | 0.646 | 0.659 | | 0.666 | 0.500 | |
| *Light-intensity physical activity* | | | | | | |
| Normal vision | Ref | Ref | 1871.5 | Ref | Ref | 1900.1 |
| Uncorrected refractive error | 41.3 (−71.8 to 154.3) | 25.1 (−86.5 to 136.8) | 1896.6 | 23.7 (−84.1 to 131.5) | 6.5 (−101.6 to 114.7) | 1906.7 |
| Visual impairment | 292.8 (−37.5 to 623.1) | 7.3 (−77.9 to 505.1) | 2085.1 | −5.7 (−256.4 to 245.0) | −15.5 (−264.5 to 233.4) | 1884.6 |
| P trend | 0.047 | 0.198 | | 0.717 | 0.953 | |
| *Lifestyle physical activity* | | | | | | |
| Normal vision | Ref | Ref | 815.9 | Ref | Ref | 654.5 |
| Uncorrected refractive error | 9.9 (−55.74 to 75.5) | −2.4 (−70.1 to 65.3) | 813.5 | −13.3 (−59.6 to 33.1) | −15.0 (−60.6 to 30.6) | 639.5 |
| Visual impairment | 115.9 (−112.8 to 344.5) | 45.9 (−157.2 to 249.1) | 861.8 | −68.6 (−187.6 to 50.3) | −81.3 (−181.7 to 19.0) | 573.2 |
| P trend | 0.330 | 0.842 | | 0.337 | 0.249 | |
| *Moderate-to-vigorous physical activity* | | | | | | |
| Normal vision | Ref | Ref | 62.6 | Ref | Ref | 49.8 |
| Uncorrected refractive error | 11.4 (−4.2 to 27.0) | 10.2 (−6.8 to 27.2) | 72.9 | 2.1 (−8.4 to 12.6) | −2.9 (−11.5 to 5.7) | 46.9 |
| Visual impairment | 2.7 (−28.3 to 33.7) | 3.9 (−25.9 to 33.7) | 66.5 | 1.7 (−29.3 to 32.7) | −2.7 (−25.5 to 20.1) | 47.1 |
| P trend | 0.226 | 0.287 | | 0.624 | 0.457 | |

*MV-adjusted models included age (continuous), BMI, race, household income, employment status, smoking status and chronic illness.
BMI, body mass index; MV, multivariate; NHANES, National Health Nutrition and Examinations Survey.

**Table 4** Accelerometer measured activity pattern (minutes per week) by vision status in the NHANES (2003–2006) adults aged 50 years and older

| | Male | | | Female | | |
|---|---|---|---|---|---|---|
| | Unadjusted β (95% CI) | MV-adjusted β (95% CI)* | Marginal mean | Unadjusted β (95% CI) | MV-adjusted β (95% CI)* | Marginal mean |
| *Sedentary time* | | | | | | |
| Normal vision | Ref | Ref | 3756.2 | Ref | Ref | 3502.5 |
| Uncorrected refractive error | −42.0 (−196.9 to 112.9) | −100.7 (−271.5 to 70.0) | 3655.5 | 211.0 (47.9 to 374.2) | 123.8 (−52.4 to 300.0) | 3626.3 |
| Visual impairment | 290.9 (−26.9 to 608.7) | 230.8 (−183.3 to 513.0) | 3921.0 | 318.3 (44.4 to 592.2) | 179.6 (−104.9 to 464.1) | 3682.1 |
| P trend | 0.244 | 0.816 | | 0.003 | 0.098 | |
| *Light-intensity physical activity* | | | | | | |
| Normal vision | Ref | Ref | 1815.6 | Ref | Ref | 1914.4 |
| Uncorrected refractive error | −202.7 (−330.4 to −75.0) | −134.9 (−257.6 to −12.2) | 1680.7 | −115.9 (−187.5 to −44.3) | −34.9 (−103.6 to 24.8) | 1875.0 |
| Visual impairment | −196.0 (−311.91 to −80.1) | −57.2 (−182.2 to 67.81) | 1758.4 | −194.8 (−335.3 to −54.2) | −56.6 (−205.9 to 92.7) | 1857.8 |
| P trend | <0.001 | 0.063 | | 0.001 | 0.234 | |
| *Lifestyle physical activity* | | | | | | |
| Normal vision | Ref | Ref | 632.2 | Ref | Ref | 507.2 |
| Uncorrected refractive error | −126.6 (−190.4 to −62.8) | −54.4 (−106.1 to −2.7) | 577.8 | −103.5 (−143.2 to −63.8) | −41.0 (−79.8 to −2.3) | 466.1 |
| Visual impairment | −224.1 (−298.9 to −149.2) | −56.5 (−140.0 to 27.0) | 575.6 | −221.0 (−269.2 to −172.8) | −82.8 (−147.8 to −17.8) | 424.4 |
| P trend | <0.001 | 0.051 | | <0.001 | 0.009 | |
| *Moderate-to-vigorous physical activity* | | | | | | |
| Normal vision | Ref | Ref | 48.1 | Ref | Ref | 30.3 |
| Uncorrected refractive error | −9.2 (−23.3 to 4.9) | −5.0 (−21.5 to 9.4) | 42.1 | −4.4 (−17.1 to 8.45) | 3.4 (−8.6 to 15.4) | 33.7 |
| Visual impairment | −16.7 (−43.7 to 10.3) | −5.0 (−36.6 to 26.7) | 43.2 | 0.2 (−21.8 to 22.3) | 13.9 (−10.6 to 38.5) | 44.2 |
| P trend | 0.127 | 0.579 | | 0.762 | 0.299 | |

*MV-adjusted models included age (continuous), BMI, race, household income, employment status, smoking status and chronic illness.
BMI, body mass index; MV, multivariate; NHANES, National Health Nutrition and Examinations Survey.

previous study has compared levels of activity in visually impaired adolescents (a combination of uncorrected refractive error and non-refractive visual impairment) to those with normal vision, and it reported similar findings. In a sample of 53 adolescents completing the International Physical Activity Questionnaire—Short Form, physical activity levels of visually impaired adolescents and sighted adolescents were similar (p>0.05).[24] Findings from the present study add to this work through using a large, population-based sample and objective measures of physical activity. Moreover, the present study is the first to compare differences in sedentary time between adolescents with impaired vision and normal sight. One plausible reason for these findings is that adolescents with normal vision have very low levels of physical activity[25]; therefore minimising any difference in a reduction of physical activity owing to a disability (eg, reduced eyesight). Another reason for the lack of association could be owing to physical education. Indeed, all adolescents regardless of visual impairment are required to partake in physical education and thus acquire similar levels of physical activity during the school day. The present study found that the estimated marginal mean of MVPA in those with non-refractive visual impairment was low, 156 min per week in males and 55 min per week in females, and sedentary time high. Interventions are needed to promote physical activity in adolescents with visual impairments.

The present finding that adults aged 50 years and older with non-refractive visual impairment accumulated lower lifestyle physical activity than those with normal vision, specifically in women, supports that of previous research and adds to it through the use of objective physical activity measures. For example, in a recent study of 6634 older English adults those with fair–poor and good eyesight were significantly more likely to be inactive, categorised by self-report, than those who reported excellent eyesight.[14] Further research is needed to understand why older adults with reduced eyesight have lower levels of physical activity. One plausible explanation may be fear of going outside, owing to falling or suffering other accidents. This low level of physical activity is of concern as this population may be at an increased risk of chronic illness, such as higher risk of cancer[24] and also have associated risk factors such as higher smoking rates,[20] independent of physical activity. Moreover, those who are visually impaired often report having a low quality of life.[26]

A high prevalence of sedentary time in those aged 20–49 years with non-refractive visual impairment should be noted. Indeed, the present study has shown higher levels of sedentary time among women with non-refractive visual impairment compared with those with normal vision. Refractory visual problems can normally be corrected and are likely to be less disabling than non-refractive visual impairment. Moreover, non-refractive visual impairment are likely to be comorbid. Therefore, those with non-refractive visual impairment may be a population with greater barriers to physical activity participation.

The lack of association between sedentary time and vision status in older adults is interesting. A plausible explanation is that as adults age, sedentary time is likely to increase regardless of disability status. Therefore, when those with and without visual impairment reach older adulthood the difference in time spent sedentary is negligible. A rationale for stronger associations with higher levels of sedentary behaviour in women but not men with visual impairment is elusive and further qualitative research to explain this finding is required.

Clear strengths of this study are the large population-based sample of US adolescents and adults and objective measurement of physical activity and sedentary time. Moreover, our statistical models controlled for a wide range of demographic and behavioural covariates. However, the data must be interpreted in light of its limitations. Analyses are of a cross-sectional design and thus it is not known whether visual impairment leads to low levels of activity and high levels of sedentary time or vice versa. Indeed, adequate levels of physical activity and lower sedentary time may decrease risk for visual impairment by reducing risk for diabetes and associated complications such as cataracts and diabetic retinopathy. The limited collected information on visual impairment (eg, lack of information on eye disease such as cataracts or macular degeneration) and the length of time one has been visually impaired may have introduced bias into the analyses. Future research should consider collecting data on specific eye conditions and length of time visual impairment has been present.

In conclusion, findings from the present study suggest generally low levels of physical activity and high levels of sedentary time in adolescents. However, activity patterns are similar between adolescents with visual impairment and those with normal vision. Adult women with non-refractive visual impairment have lower levels of lifestyle physical activity (aged 50+years) and higher levels of sedentary time (aged 20–49 years) than those with normal vision. Taken together, these findings highlight the need for interventions to promote physical activity and reduce sedentary time in adult populations with visual impairment, specifically adult women.

**Author affiliations**
[1]The Cambridge Centre for Sport and Exercise Sciences, Anglia Ruskin University, Cambridge, UK
[2]Deaprtment of Behavioural Science and Health, University College London, London, UK
[3]Vision and Eye Research Unit (VERU), School of Medicine, Anglia Ruskin University, Chelmsford, Essex, UK
[4]Faculty of Sport Sciences, University of Murcia, Murcia, Spain
[5]Physical Education, Zhejiang University, Hangzhou, Zhejiang, China
[6]Division of Public Health Sciences, Department of Surgery, Washington University School of Medicine, St Louis, USA
[7]Centre for Public Health Systems Science, Brown School Washington University, St Louis, USA
[8]Department of Rehabilitation Sciences, KU Leuven, University of Leuven, Leuven, Belgium
[9]University Psychiatric Centre, KU Leuven, University of Leuven, Leuven, Belgium
[10]Research and Development Unit, Parc Sanitari Sant Joan de Deu, Barcelona, Spain
[11]Department of Psychological Medicine, Institute of Psychiatry, King's College London, London, UK

$^{12}$NICM Health Research Institute, University of Western Sydney, Sydney, Australia
$^{13}$Department of Epidemiology, Medical University of Vienna, Austria, Vienna, Austria

**Correction notice** This article has been corrected since it was published. The correct licence for the paper is CC-BY.

**Contributors** LY did the analyses. LS and SEJ interpreted the results and drafted the manuscript. LS, SEJ, SP, GFL-S, LH, CC, DV, AK, BS, JF, and LY critically appraised the manuscript and approved the final version before submission.

**Funding** SJ is supported by Cancer Research UK (C1417/A22962). GFLS is supported by a postdoctoral fellowship from the Seneca Foundation—Agency for Science and Technology of the Region of Murcia, Spain (20390/PD/17).

**Competing interests** None declared.

**Patient consent for publication** Obtained.

**Ethics approval** The NHANES obtained ethical approval from the National Centre for Health Statistics Research Ethics Review Board.

**Provenance and peer review** Not commissioned; externally peer reviewed.

**Data sharing statement** All data relevant to the study are included in the article or uploaded as supplementary information.

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
