## [Reviewer comments · BMJ Open]

ARTICLE DETAILS

TITLE (PROVISIONAL)	Visual impairment and objectively measured physical activity and sedentary behavior in US adolescents and adults: a cross-sectional study
AUTHORS	Smith, Lee; Jackson, Sarah; Pardhan, Shahina; López-Sánchez, Guillermo Felipe; Hu, Liang; Chao, Cao; Vancampfort, D.; Koyanagi, Ai; Stubbs, B; Firth, Joseph; Yang, Lin

VERSION 1 - REVIEW

REVIEWER	Cindy Sit The Chinese University of Hong Kong Hong Kong
REVIEW RETURNED	12-Nov-2018

GENERAL COMMENTS	This paper examines the levels of physical activity (PA) and sedentary time (ST) in a representative sample of US adolescents and adults with and without visual impairment. It has strengths such as involving a large population-based sample, using objective measure of PA and ST, and examining the influence of visual impairment on PA and ST. However, there are several issues that require attention. 1. Given the sample includes both adolescents and adults with and without visual impairment, it would be better to provide relevant literature on their PA and ST, especially how visual impairment impacts such behaviours. Also, why visual impairment is an area of interest? More justifications are needed.2. It appears that the age range of participants is between 12 and 85 years? Age group classifications are unclear. PA and ST are very different among adolescents, young, middle-aged and older adults. Evidence indicates that adolescents' PA is more spontaneous and intermittent in nature (and of higher intensity) and that using the epoch length 1 min may not be able to capture their PA bouts. Additionally, would the standard cpm cutoff be suitable for use in such wide age groups?3. It is not sure if the second age group (20 years and above) is sufficient to reflect the characteristics of participants in the study, though age was adjusted in the generalized linear models. It might be interesting to have the third (middle-aged) and even the fourth (old-aged) groups. Visual impairment or disability commonly exists due to aging.4. Lifestyle (e.g., consuming alcohol) and diet factors are not included. Demographic information such as occupation (e.g., blue
---

	vs. white collar, full-time vs. part-time, shift duty or non-shift duty) could also be a potential confounder. 5. Discussions could be strengthened by addressing the issues mentioned above.
--	---

REVIEWER	Elizabeth Lenz The College at Brockport-SUNY, USA
REVIEW RETURNED	29-Nov-2018

GENERAL COMMENTS	The manuscript entitled, “ Visual impairment and objectively measured physical activity and sedentary behavior in US adolescents and adults” examined the relationship between physical activity and sedentary behavior time between individuals with and without visual impairments. This study was unique in that it used data from NHANES and objective measures of activity and sedentary time as well as objective examination of visual acuity. The researchers observed no differences in adolescent physical activity and sedentary time regardless of vision, however differences were seen in adults especially regarding sedentary behavior and lifestyle physical activity in women. This information supports the few self-reported physical activity studies conducted within this population as well as adds to the literature because of the strength in methodology. The following are suggestions for improving the wording of some sentences in the manuscript. I did not find any significant errors or concerns with this paper. Abstract. 1. The abstract is very well written and includes all necessary pieces but it is missing an overall conclusion. It currently ends with results. I suggest including a summary statement. Strengths and Weaknesses Line 71. This bullet needs to be edited. Maybe reword to something like “Risk for developing diabetes and associated complications such as cataracts and diabetic retinopathy may be reduced in those with visual impairments by participating in adequate levels of PA and lower amounts of SB.” Introduction Line 108. I suggest removing the word “that” and spelling out United Kingdom Line 111. I suggest removing “We” and use “Researchers” or something similar Line 119. I suggest adding a reference to your statement about accuracy with accelerometry and its use in those with visual impairments Line 119-121. The sentence beginning with “moreover” may not be needed in the introduction and could be used in the discussion? I am not finding it relevant to the purpose of your study – comparing PA and SB in those with and without visual impairment in the UK. Methods Study Pop. Line 155. Remove the word “We” and say something along the lines of “The following data was extracted or Researchers extracted data....” Line 158. Same concept with “We” Accelerometer
--

	Line 184. You could remove “count” so the Actigraph recorded data for PA in the form of counts. It is less repetitive. Line 195 – I think you meant vigorous instead of vigour? Just trying to remain consistent with the other times you used the word. Overweight/Obese Line 207-208. You abbreviate mobile exam. Center throughout the paper except in this sentence. Line 216-217. You may consider rewording to “if they self-reported being told by a physician...” Statistical Analysis As mentioned earlier please remove the word “We” – Lines 225, 227, 237 Results Line 243. Please consider removing the word “Our” and using “The study population....” Please remove “We” from lines 269, 273, 276 Discussion Line 306. Consider removing the word “that”
--	---

VERSION 1 – AUTHOR RESPONSE

Reviewer(s)' Comments to Author:

Reviewer: 1

Reviewer Name: Cindy Sit

Institution and Country: The Chinese University of Hong Kong

Hong Kong

Please state any competing interests or state ‘None declared’: None declared.

Please leave your comments for the authors below

This paper examines the levels of physical activity (PA) and sedentary time (ST) in a representative sample of US adolescents and adults with and without visual impairment. It has strengths such as involving a large population-based sample, using objective measure of PA and ST, and examining the influence of visual impairment on PA and ST. However, there are several issues that require attention.

1. Given the sample includes both adolescents and adults with and without visual impairment, it would be better to provide relevant literature on their PA and ST, especially how visual impairment impacts such behaviours. Also, why visual impairment is an area of interest? More justifications are needed.

Response: Thank you we have now expanded and included more on the points raised above throughout the introduction. See below:

“Persons with disabilities have been shown to have low levels of physical activity and high sedentary behaviour levels and understanding differences versus the general population is important.¹¹ The disability of reduced eyesight may be a key barrier to an active lifestyle in adults and adolescents. It has been suggested that in people with visual impairment there is a lack of access to recreational and athletic programmes, and help or encouragement in developing suitable and safe physical recreation skills and habits.¹² Moreover, this population experiences activity limitations in walking, and

environmental barriers such as transport and lack of accessible exercise equipment can hamper a person's ability to be physically active.¹²⁻¹³ The authors of the present paper have shown in a sample of 6634 United Kingdom participants (mean age 65.0±9.2 years) those with poor vision were twice as likely to be physically inactive than those with good eyesight. Similar findings were found for the variable 'recognition of friends across street' and 'reading ordinary newspaper'.¹⁴ The present authors have also found similar associations in young people.¹⁵ These findings are of importance owing to a high prevalence of reduced eyesight. For example, in the USA it has been estimated that approximately 14 million individuals aged 12 years or older have visual impairment (defined as distance visual acuity of 20/50 or worse).¹⁶

2. It appears that the age range of participants is between 12 and 85 years? Age group classifications are unclear. PA and ST are very different among adolescents, young, middle-aged and older adults. Evidence indicates that adolescents' PA is more spontaneous and intermittent in nature (and of higher intensity) and that using the epoch length 1 min may not be able to capture their PA bouts. Additionally, would the standard cpm cutoff be suitable for use in such wide age groups?

Response: We acknowledge that physical activity and sedentary behavior are very different among adolescents, young, middle-aged and older adults. For instance adolescents' PA are of higher intensity, therefore we analyzed light intensity and lifestyle PA among adults only, but not those younger than 20 years old.

We have used the derived variables from NHANES accelerometer study that have used standard cut-off by NCI program (https://epi.grants.cancer.gov/nhanes_pam/) and been widely used in published study (https://journals.sagepub.com/doi/abs/10.1177/0890117116684241?rfr_dat=cr_pub%3Dpubmed&url_ver=Z39.88-2003&rfr_id=ori%3Arid%3Acrossref.org&journalCode=ahpa).

3. It is not sure if the second age group (20 years and above) is sufficient to reflect the characteristics of participants in the study, though age was adjusted in the generalized linear models. It might be interesting to have the third (middle-aged) and even the fourth (old-aged) groups. Visual impairment or disability commonly exists due to aging.

Response: Thanks for the comments. We have now stratified adults (20 years and above) to two sub-groups: 20-49 years, and 50 years and older. We have updated tables, results, and discussion accordingly.

4. Lifestyle (e.g., consuming alcohol) and diet factors are not included. Demographic information such as occupation (e.g., blue vs. white collar, full-time vs. part-time, shift duty or non-shift duty) could also be a potential confounder.

Response: Thank you raising this concern. We have included and adjusted for employment status as a potential confounder in the analyses. Due to the nature of collected data, information is limited to derive more refined categories for occupation information. We have included a binary variable: employed vs. unemployed in the multivariable models. With respect to dietary factors or alcohol consumption. We agree that those variables are important covariates to consider for analyzing association of physical activity with health outcomes. We however do not believe they qualify potential confounding factors in current analyses, as physical activity is the outcome.

5. Discussions could be strengthened by addressing the issues mentioned above.

Reviewer: 2

Reviewer Name: Elizabeth Lenz

Institution and Country: The College at Brockport-SUNY, USA

Please state any competing interests or state 'None declared': None Declared

Please leave your comments for the authors below

The manuscript entitled, " Visual impairment and objectively measured physical activity and sedentary behavior in US adolescents and adults" examined the relationship between physical activity and sedentary behavior time between individuals with and without visual impairments. This study was unique in that it used data from NHANES and objective measures of activity and sedentary time as well as objective examination of visual acuity. The researchers observed no differences in adolescent physical activity and sedentary time regardless of vision, however differences were seen in adults especially regarding sedentary behavior and lifestyle physical activity in women. This information supports the few self-reported physical activity studies conducted within this population as well as adds to the literature because of the strength in methodology. The following are suggestions for improving the wording of some sentences in the manuscript. I did not find any significant errors or concerns with this paper.

Abstract.

1. The abstract is very well written and includes all necessary pieces but it is missing an overall conclusion. It currently ends with results. I suggest including a summary statement.

Response: We have now included a summary statement in the abstract.

"Taken together, these findings highlight the need for interventions to promote physical activity and reduce sedentary time in adult populations with visual impairment, specifically adult women."

Strengths and Weaknesses

Line 71. This bullet needs to be edited. Maybe reword to something like "Risk for developing diabetes and associated complications such as cataracts and diabetic retinopathy may be reduced in those with visual impairments by participating in adequate levels of PA and lower amounts of SB."

Response: Thank you, we have now corrected as suggested.

Introduction

Line 108. I suggest removing the word "that" and spelling out United Kingdom

Response: We have removed the word "that" and spelled out United Kingdom.

Line 111. I suggest removing "We" and use "Researchers" or something similar

Response: We have removed the word "we".

Line 119. I suggest adding a reference to your statement about accuracy with accelerometry and its use in those with visual impairments

Response: Thank you, we have now added a reference relating to the accuracy of accelerometer. However, there is no existing literature on the accuracy of accelerometers in those with visual impairment. We do not see any plausible reason as to why the devices accuracy would differ between the visually impaired and those with "normal" vision.

Line 119-121. The sentence beginning with "moreover" may not be needed in the introduction and could be used in the discussion? I am not finding it relevant to the purpose of your study – comparing PA and SB in those with and without visual impairment in the UK.

Response: We have removed this sentence.

Methods

Study Pop.

Line 155. Remove the word “We” and say something along the lines of “The following data was extracted or Researchers extracted data....”

Response: We have rephrased the sentence accordingly.

Line 158. Same concept with “We”

Response: We have rephrased the sentence accordingly.

Accelerometer

Line 184. You could remove “count” so the Actigraph recorded data for PA in the form of counts. It is less repetitive.

Response: We have removed “count”.

Line 195 – I think you meant vigorous instead of vigour? Just trying to remain consistent with the other times you used the word.

Response: We have corrected “vigour” to “vigorous”.

Overweight/Obese

Line 207-208. You abbreviate mobile exam. Center throughout the paper except in this sentence.

Response: We have revised it to MEC to be consistent throughout the paper.

Line 216-217. You may consider rewording to “if they self-reported being told by a physician...”

Response: We have reworded it to “if they self-reported being told by a physician...”

Statistical Analysis

As mentioned earlier please remove the word “We” – Lines 225, 227, 237

Response: We have removed the word “We” in the text.

Results

Line 243. Please consider removing the word “Our” and using “The study population....”

Please remove “We” from lines 269, 273, 276

Response: We have removed the word “We” in the text and rephrased “Our”.

Discussion

Line 306. Consider removing the word “that”

Response: We have removed the word “that”.